# Continuous Prediction Model of Carbon Content in 120 t Converter Blowing Process

**Dazhi Wang, Fang Gao, Lidong Xing, Jianhua Chu and Yanping Bao ***

State Key Lab of Advanced Metallurgy, University of Science & Technology Beijing, Beijing 100083, China; b20180528@xs.ustb.edu.cn (D.W.); d202110626@xs.ustb.edu.cn (F.G.); b20170513@xs.ustb.edu.cn (L.X.); b20180527@xs.ustb.edu.cn (J.C.)
* Correspondence: baoyp@ustb.edu.cn; Tel.: +86-138-1076-8855

**Abstract:** A continuous prediction model of carbon content of 120 t BOF is established in this paper. Based on the three-stage decarburization theory and combined with the production process of 120 t converter, the effects of oxygen lance height and top blowing oxygen flow rate are also considered in the model. The explicit finite difference method is used to realize continuous prediction of carbon content in the converter blowing process. The model parameters such as ultimate carbon content in molten pool are calculated according to the actual data of 120 t BOF, which improves the hit rate of the model. Process verification and end-point verification for the continuous prediction model have been carried out, and the results of process verification indicate that the continuous prediction model established in the paper basically accords with the actual behavior of decarburization. Moreover, the hit ratio of the continuous prediction model reached 85% for the prediction of end-point carbon content within a tolerance of ±0.02%.

**Keywords:** continuous prediction model; carbon content; three-stage decarburization theory; 120 t BOF; the explicit finite difference method



## 1. Introduction

Converter steelmaking is the main steelmaking production mode in China. Due to the limitation of scrap shortage, converter steelmaking will be dominant for a long time [1]. Oxygen converter mainly blows high-pressure and high-purity oxygen from the oxygen lance on the top of the converter to reduce the content of carbon, phosphorus, and other elements in hot metal and increase the temperature, so as to obtain molten steel with composition and temperature meeting the target requirements [2,3]. One of the important links of converter steelmaking is to realize the accurate control of blowing end-point. Most converter steelmaking production mainly depends on manual experience or sub-lance detection to achieve end-point control in China [4,5]. Depending on the experience of production personnel to control composition and temperature, the accuracy of end-point hit is low. The sub-lance can only detect intermittently in the whole process of blowing. Therefore, the establishment of appropriate converter prediction and control model has attracted more and more attention of iron and steel enterprises.

The end-point control of converter generally includes end-point temperature control and end-point carbon content control. Due to the fast decarburization speed in converter and the narrow range of carbon content required by steel grade specification, it is very difficult to predict and control the end-point carbon content [6,7]. Accurately controlling the end-point carbon content can not only avoid molten steel peroxidation and reduce alloy burning loss after furnace, but also reduce carbon emission in steelmaking process to a certain extent. Mechanism model [8,9], statistical model [10,11], and neural network model [12,13] are commonly used to predict the carbon content of converter.

There are complex physical and chemical reactions in steelmaking process. A simple statistical model is difficult to apply to converter smelting with so many influencing factors.

In recent years, researchers have widely used neural network modeling methods to obtain high prediction accuracy [14–18]. Neural network model needs a large number of complete and accurate data samples for training in order to ensure the prediction accuracy. For many iron and steel enterprises in China, it is difficult to provide qualified samples for model training due to the lack of necessary data acquisition and storage systems. Moreover, the statistical model and neural network model are difficult to realize the continuous prediction of carbon content in molten pool. The off-gas analysis model uses the information of CO and $CO_2$ in the off-gas to calculate the carbon content in the bath, so as to realize the dynamic prediction of the blowing process [19]. Liu et al. corrected the exponential model parameters by means of isometric multi-point continuous correction [20]. Meyer and Glasgow et al. established an exponential decay model [21]. However, the off-gas analysis model needs off-gas analysis equipment, which many iron and steel enterprises do not have.

The mechanism model can well predict the process of carbon oxidation in the furnace. For the converter with stable raw material conditions and relatively fixed operation mode, it can ensure better prediction accuracy. Based on the three-stage decarburization principle and combined with the specific production process of 120 t converter in an iron and steel enterprise in China, the continuous prediction model of carbon content considering the influence of oxygen lance height and oxygen flow rate is established. The model parameters such as ultimate carbon content of molten steel are calculated according to the enterprise data. The dynamic transmission of process parameters is realized by finite difference method. This model only needs raw material information and process information to run without additional testing information. This model is more practical for many iron and steel enterprises lacking testing equipment.

## 2. Description of Decarburization Process of Converter

In the process of converter blowing, a series of physical and chemical reactions occur between hot metal and high-speed oxygen jet ejected from the top oxygen lance. Three reaction zones are mainly formed in the furnace: oxygen jet impact reaction zone, slag-metal interface reaction zone, and emulsion phase reaction zone. Decarburization mainly occurs in the oxygen jet impact reaction zone.

Equations (1) and (2) describe the decarburization reaction in the oxygen jet reaction zone.

$$[C] + \frac{1}{2}O_2 = CO \tag{1}$$

$$CO + \frac{1}{2}O_2 = CO_2 \tag{2}$$

In addition to the oxidation of carbon, the following reactions occur in the converter. The oxidation of Si, Mn, and Fe as shown in Equations (4)–(6) occurs in the oxygen jet reaction zone. The reaction shown in Equations (7)–(10) occurs in the slag–metal interface reaction zone. The reaction shown in Equation (3) occurs in the emulsion phase reaction zone.

$$[C] + (FeO) = CO + [Fe] \tag{3}$$

$$[Si] + O_2 = (SiO_2) \tag{4}$$

$$[Mn] + \frac{1}{2}O_2 = (MnO) \tag{5}$$

$$[Fe] + \frac{1}{2}O_2 = (FeO) \tag{6}$$

$$2[P] + 5(FeO) = 5[Fe] + (P_2O_5) \tag{7}$$

$$4(CaO) + (P_2O_5) = (4CaO \cdot P_2O_5) \tag{8}$$

$$2(CaO) + (SiO_2) = (2CaO \cdot SiO_2) \tag{9}$$

$$[\text{Si}] + 2(\text{FeO}) = (\text{SiO}_2) + 2[\text{Fe}] \tag{10}$$

$$[\text{Mn}] + (\text{FeO}) = (\text{MnO}) + [\text{Fe}] \tag{11}$$

From the thermodynamic point of view, the ability of many chemical elements in the molten pool, especially silicon and manganese, to combine with oxygen is much greater than that of carbon. When $w(\text{Si}) + 0.25\,w(\text{Mn}) < 0.1\%$, the content of silicon and manganese has little effect on the rate of carbon oxygen reaction.

It is generally believed that the decarburization process of converter can be divided into three stages [22]. As shown in Figure 1, in the early stage, due to the high content of Si and Mn and low temperature, Si and Mn are oxidized first. Oxygen is more consumed in the oxidation of Si and Mn. In the middle stage of blowing, the reaction of Si and Mn is close to the end. The temperature of molten pool is high, and the transfer rate of carbon in liquid steel to the reaction interface is very large. Oxygen is basically completely consumed in the decarburization reaction. At this stage, the decarburization reaction rate reaches the maximum and remains basically unchanged. In the later stage of decarburization reaction, due to the reduction of carbon content in the molten pool, the decarburization reaction began to be limited by carbon mass transfer. The decarburization rate is related to the carbon content and begins to decrease.

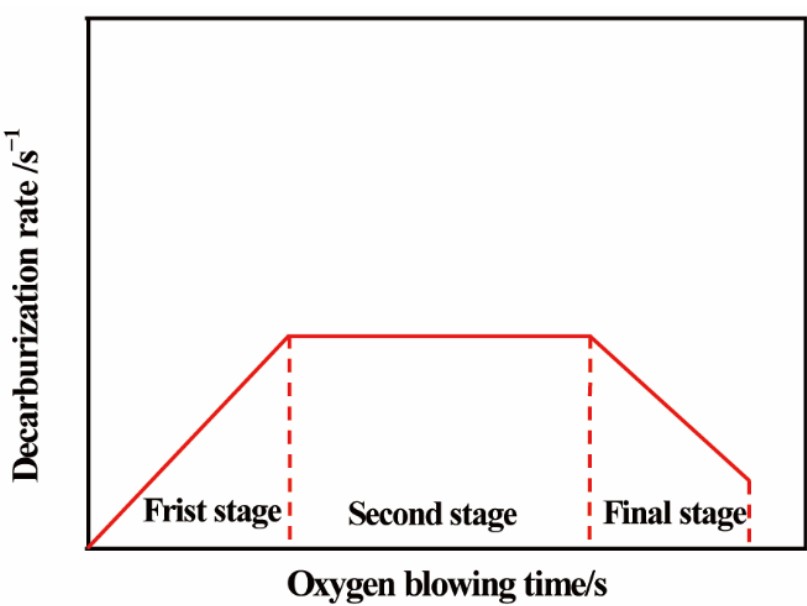

**Figure 1.** Traditional three-stage decarburization theory for BOF process.

Generally, three conditions are required for the converter to finish smelting. The phosphorus content of molten steel is lower than the requirements of steel grade. The carbon content and temperature of molten steel meet the process requirements. Since it is difficult to reduce the phosphorus content of molten steel in the subsequent refining process, the requirements for phosphorus content at the end of converter are stricter. However, the control of end-point carbon content is equally important. Too high carbon content will increase the content of sulfur and phosphorus, affect the desulfurization and dephosphorization operation, and too low carbon content will increase the content of oxygen and nitrogen. The control of end-point carbon content has an important impact on molten steel quality and smelting efficiency.

## 3. Brief Introduction of 120 t Converter Production Process

Based on the production process of 120 t converter in an iron and steel enterprise, the modeling is carried out in this paper. The hot metal conditions of the 120 t converter are shown in Figure 2. As shown in Figure 2a, the temperature of hot metal is distributed between 1305 °C and 1395 °C. Among them, the hot metal temperature of 87% of the

heats is concentrated at 1320–1365 °C. The temperature fluctuation of hot metal is small. As shown in Figure 2b, the carbon content of hot metal is distributed between 3.95% and 4.35%. The carbon content of hot metal of 75% of the heats is concentrated at 4.15–4.25%. The carbon content of hot metal is relatively stable.

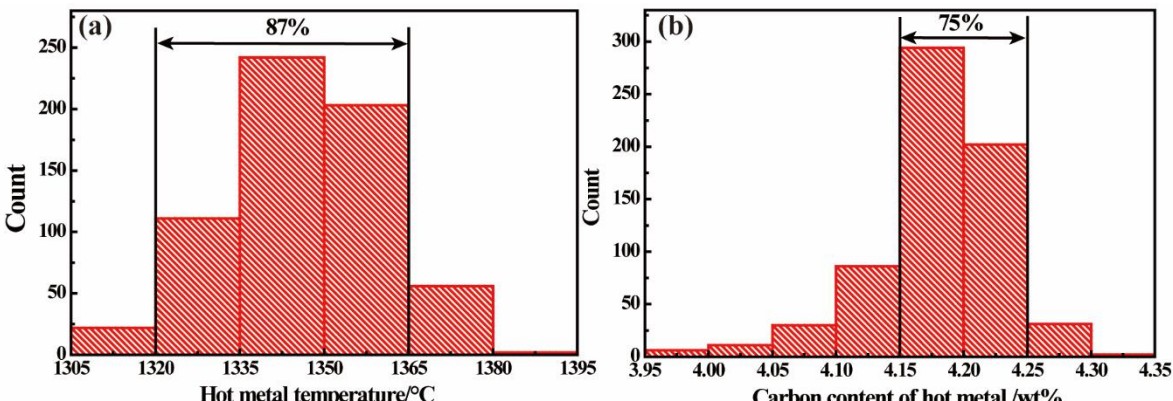

**Figure 2.** Hot metal conditions of the 120 t converter. (**a**) Distribution of hot metal temperature. (**b**) Distribution of carbon content of hot meta.

Figure 3 shows the oxygen blowing process of the 120 t converter. During the blowing process, the production is carried out in strict accordance with the specified oxygen lance height curve, unless splashing and other accidents occur. The oxygen blowing flow is also implemented in strict accordance with the specified process.

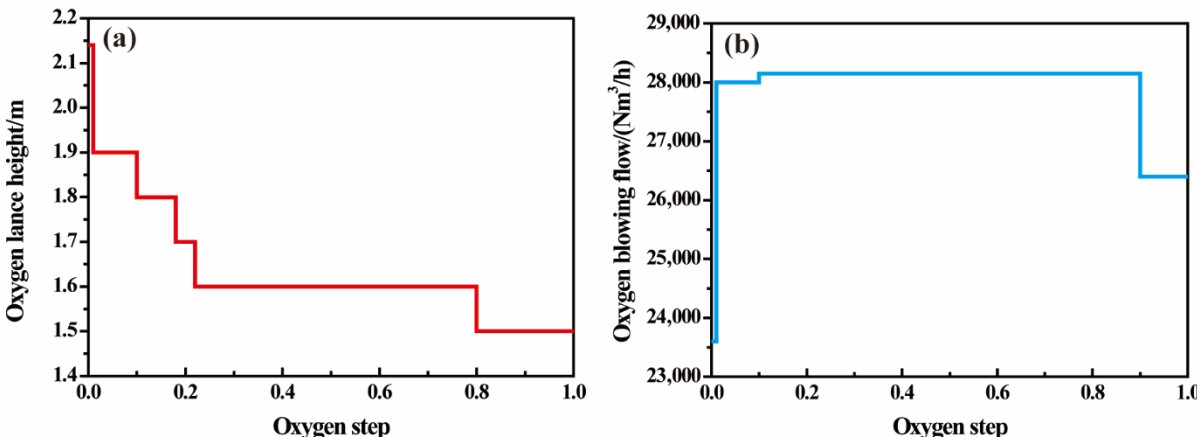

**Figure 3.** The oxygen blowing process of the 120 t converter. (**a**) Oxygen lance height control curve. (**b**) Oxygen blowing flow control curve.

The carbon content and temperature control at the end of oxygen blowing are shown in Figure 4. As shown in Figure 4a, the end-point temperature of molten steel is distributed between 1590 and 1670 °C. The end-point molten steel temperature of 90% of heats is concentrated at 1610–1650 °C. As shown in Figure 4b, the end-point carbon content of molten steel is distributed between 0.03 and 0.13%. The end-point molten steel carbon content of 91% of the heats is concentrated at 0.03–0.09%. The end-point molten steel temperature and carbon content of converter are controlled stably.

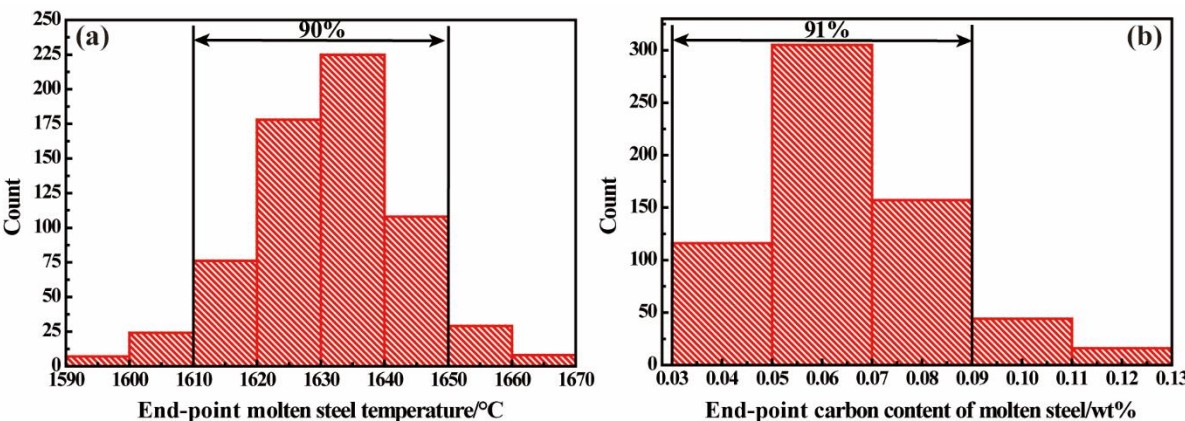

**Figure 4.** End-point molten steel conditions of 120 t converter.

It can be seen that the hot metal composition fluctuation of the iron and steel enterprise is small and the production process is stable, which is beneficial to establish the carbon content prediction model.

## 4. Establishment of the Continuous Prediction Model of Carbon Content
### 4.1. Structure of the Continuous Prediction Model of Carbon Content

According to the classical three-stage decarburization theory, in the first stage, the decarburization rate increases linearly with time. At this stage, the decarburization rate is mainly affected by the oxidation rate of Si and Mn.

The first stage of decarburization rate can be expressed as

$$-\frac{dC}{dt} = W_{steel}k_1 t \tag{12}$$

The decarburization rate of the second stage is constant. Decarburization rate is related to total oxygen entering molten pool per unit time. The second stage of decarburization rate can be expressed as

$$-\frac{dC}{dt} = W_{steel}\alpha v_{Oxygen} \tag{13}$$

The decarburization rate in the third stage is related to the carbon content. The third stage of decarburization rate can be expressed as

$$-\frac{dC}{dt} = W_{steel}k_3\left(C_{[C]} - C_0\right) \tag{14}$$

where, $-dC/dt$ is the decarburization rate, $kg \cdot s^{-1}$; t is the time, s; $C_{[C]}$ is the carbon content, wt%; $C_0$ is the ultimate carbon content, wt%; and $k_i$ (i = 1, 2, 3) is the constant coefficient; $W_{steel}$ is the weight of molten steel in converter bath, kg; $\alpha$ is the decarburization oxygen efficiency (carbon content that can be removed by 1 kg oxygen), wt%/kg; $v_{Oxygen}$ is the amount of oxygen entering the molten pool per unit time, kg/s.

The classical three-stage decarburization theory does not consider the effect of oxygen blowing process on decarburization reaction. In steelmaking process, oxygen blowing process has a great influence on decarburization. If the oxygen lance height is low, the contact area between carbon and oxygen in molten steel is large. The carbon content will decrease at a faster rate. The oxygen lance height and oxygen flow rate of the 120 t converter described in this paper will change during the blowing process. Li et al. [23] found that the effect of oxygen lance height and oxygen flow rate on decarburization rate is basically in linear proportion. According to this discovery, the decarburization index model in the later stage of converter is improved and the accuracy of the model is effectively improved.

The decarburization model considering the influence of oxygen blowing process can be expressed as:

The first stage of decarburization rate can be expressed as

$$-\frac{dC}{dt} = W_{steel}\beta\frac{h_{min}}{h}\frac{Q_{top}}{Q_{topmax}}k_1 t \tag{15}$$

The second stage of decarburization rate can be expressed as

$$-\frac{dC}{dt} = W_{steel}\beta\frac{h_{min}}{h}\frac{Q_{top}}{Q_{topmax}}\alpha v_{Oxygen} \tag{16}$$

The third stage of decarburization rate can be expressed as

$$-\frac{dC}{dt} = W_{steel}\beta\frac{h_{min}}{h}\frac{Q_{top}}{Q_{topmax}}k_3\left(C_{[C]} - C_0\right) \tag{17}$$

where, $\beta$ is the constant coefficient; h is the oxygen lance height, m; $h_{min}$ is the lowest oxygen lance height, m; $Q_{top}$ is the oxygen flow rate, $Nm^3/h$; $Q_{topmax}$ is the maximum oxygen flow rate, $Nm^3/h$.

*4.2. Effect of Bath Temperature on Decarburization*

In addition to the influence of oxygen blowing process, the molten pool temperature is also one of the important factors affecting the decarburization rate. Especially in the third stage of decarburization, the effect of temperature is more obvious. When the temperature increases, the decarburization rate will be faster. The relationship between decarburization rate constant coefficient and temperature derived from Arrhenius equation is as follows

$$k_i = \mu_i e^{-\frac{E_a}{RT}} \tag{18}$$

where $\mu_i$ is the constant coefficient; $E_a$ is the reaction activity; R is the gas constant; T is the bath temperature.

The bath temperature is calculated by the heat balance of converter bath.

$$T = \frac{Q_{hot\,metal} + Q_{reaction} + Q_{dust}^{rh} - Q_{dust}^{ph} - Q_{gas}^{ph} - Q_{iron-bead}^{ph} - Q_{splash}^{ph} - Q_{scrap}^{melt} - Q_{loss}}{c_{molten\,steel} \times W_{molten\,steel} + c_{slag} \times W_{slag}} \tag{19}$$

where $Q_{hot\,metal}$ is the physical heat of hot metal; $Q_{reation}$ is the heat of chemical oxidation of hot metal elements; $Q_{dust}^{rh}$ is the heat of smoke dust oxidation; $Q_{dust}^{ph}$ is the physical heat of smoke dust; $Q_{gas}^{ph}$ is the physical heat of furnace gas; $Q_{iron-bead}^{ph}$ is the physical heat of iron bead; $Q_{splash}^{ph}$ is the physical heat of splash; $Q_{scrap}^{melt}$ is the heat of scrap melting; $Q_{loss}$ is the converter heat loss; $c_{molten\,steel}$ and $c_{slag}$ are the specific heat capacities of molten steel and slag, respectively; $W_{molten\,steel}$ and $W_{slag}$ are the weights of molten steel and slag, respectively.

*4.3. Differentiation of Three Decarburization Stages*

Equations (15)–(17) in this paper describe the decarburization behavior in each stage of the converting process. In order to completely calculate the carbon content in the converter blowing process, the link points of the three stages need to be determined.

(1) The turning point of between the first stage and the second stage of decarburization.

The main difference between the first stage and the second stage is that in the first stage, the content of Si and Mn in hot metal is high, especially the oxidation of Si inhibits the decarburization reaction to a certain extent. In this paper, the duration of the first stage

of decarburization is judged by the degree of desiliconization reaction. The desiliconization reaction of converter mainly occurs in the gas-metal reaction zone and slag-metal reaction zone [24,25], and the desiliconization reaction rate can be expressed as

$$-\frac{dSi}{dt} = A_{iz}k_{gm}\rho_m((Si\%) - (Si\%)_{sm})/100 + A_{sm}k_{sm}\rho_m((Si\%) - (Si\%)_{sm})/100 \quad (20)$$

where $-dSi/dt$ is the desiliconization reaction rate, $kg \cdot s^{-1}$; $A_{iz}$ is the interfacial area of gas-metal reaction, $m^2$; $k_{gm}$ is the mass transfer coefficient at gas-metal interface, m/s; $A_{sm}$ is the interfacial area of slag-metal reaction, $m^2$; $k_{sm}$ is the mass transfer coefficient at slag-metal interface, m/s; $\rho_m$ is the density of the bulk metal, 7000 $kg/m^3$; (Si%) is the concentration of Si in hot metal, wt%; $(Si\%)_{sm}$ is the interface concentration of Si at the slag-metal interface, wt%; $(Si\%)_{gm}$ is the interface concentration of Si at the gas-metal interface, wt%. The interface concentration of Si is 0.25 (Si%).

The interfacial area of gas-metal reaction can be expressed as

$$A_{iz} = n_n \frac{\pi r_{iz}^4}{6h_{iz}^2}\left[\left(1 + \frac{4h_{iz}^2}{r_{iz}^2}\right)^{\frac{3}{2}} - 1\right] \quad (21)$$

where $n_n$ is the number of nozzles; $r_{iz}$ is the radius of the gas-metal reaction zone, $m^2$; $h_{iz}$ is the height of the gas-metal reaction zone, m.

$h_{iz}$ and $r_{iz}$ can be expressed as

$$h_{iz} = 4.469 \times M_h^{0.66} h \quad (22)$$

$$r_{iz} = 0.5 \times 2.813 h M_d^{0.282} \quad (23)$$

$$M_h = \frac{m_n \cos(nangle)}{\rho_m g L_h^3} \quad (24)$$

$$M_d = \frac{m_t(1 + \sin(nangle))}{\rho_m g L_h^3} \quad (25)$$

$$m_n = \frac{m_t}{n_n} \quad (26)$$

$$m_t = 0.7854 \times 10^5 \times n_n d_{th}^2 P_a\left(\frac{1.27 \times P_0}{P_a} - 1\right) \quad (27)$$

where $d_{th}$ is the throat diameter of the lance, m; Pa is the ambient pressure, $P_a$; $P_0$ is the top supply pressure, $P_a$; $m_t$ is the total momentum flow rate; $m_n$ is the momentum flow rate of each nozzle; $M_h$ and $M_d$ are the dimensionless momentum flow rate.

The interfacial area of slag-metal reaction can be expressed as

$$A_{sm} = \pi \times \left(\frac{D_b^2}{4} - n_n r_{iz}^2\right) \quad (28)$$

where $D_b$ is the diameter of the bath surface, m.

The Si content in hot metal during converter blowing process can be calculated by Equation (20). At the end of the early stage of converter blowing, the Si content is usually reduced to less than 0.03% [24]. Therefore, when the Si content in hot metal is reduced to 0.03%, it is considered as the end of the first stage of decarburization.

(2) The turning point of between the second stage and the third stage of decarburization.

During the final stage of decarburization, the decarburization rate will be affected by carbon content because of the limitation of carbon mass transfer. There is a critical carbon content $C_t$, which means that the decarburization rate begins to be affected by the carbon content. When the carbon content of molten steel is lower than the critical carbon content,

the decarburization reaction enters the third stage. Generally, the critical carbon content is usually related to the stirring strength of the molten pool. The critical carbon content $C_t$ is between 0.2 and 0.6% [26]. Some scholars choose the critical carbon content as 0.35% or 0.43% [27]. In this paper, the critical carbon content $C_t$ is chosen as 0.36%. When the carbon content is 0.36%, the decarburization rates of the second decarburization stage and the third decarburization stage are equal.

The selective oxidation reaction between C and Fe at the end of converter blowing is shown in Equation (29).

$$[C] + (FeO) = CO + [Fe] \lg \frac{a_{Fe} \times \frac{P_{CO}}{P^\theta}}{a_{FeO} \times a_c} = \lg \frac{a_{Fe} \times \frac{P_{CO}}{P^\theta}}{a_{FeO} \times f_C \times w_{[C]}} = -\frac{\Delta_r G_m^0}{2.303 \times RT} = \frac{-5168}{T} + 4.741 \qquad (29)$$

where $a_C$, $a_{Fe}$, and $a_{FeO}$ are the activities of C, Fe, and FeO respectively; $f_C$ is the activity coefficient of C; $P_{CO}/P^\theta$ is the partial pressure of CO. T is the reaction temperature, K.

At the end of converter blowing, it is considered that the reaction is close to equilibrium. $f_C \approx 1$, $P_{CO}/P^\theta \approx 1$, $a_{Fe} = 1$. Equation (29) can be simplified to Equation (30).

$$\lg \frac{1}{a_{FeO} \times w_{[C]}} = \frac{-5168}{T} + 4.741 \qquad (30)$$

Table 1 shows the actual final slag composition of 120 t converter. The activities of FeO can be calculated by slag molecular theory combined with the data in Table 1.

$$a_{FeO} = \frac{n_{FeO}}{n_i} = 0.317 \qquad (31)$$

where $n_{FeO}$ is the mole number of FeO in the slag; $n_i$ is the sum of moles of free oxide and complex oxide in slag.

**Table 1.** Final slag composition of 120 t converter.

| Slag Component | CaO | MgO | MnO | SiO$_2$ | P$_2$O$_5$ | FeO |
|---|---|---|---|---|---|---|
| Average value/wt% | 45.56 | 8.76 | 3.28 | 12.93 | 3.21 | 23.00 |

The calculated value of $a_{FeO}$ is substituted into Equation (30). The relationship between the critical C content of selective oxidation of C and Fe at the end of converter blowing and the molten steel temperature is obtained, as shown in Equation (32).

$$C_0 = w_{[C]} = \frac{1}{10^{\frac{-5168}{T} + 4.741} \times 0.317} \qquad (32)$$

The average temperature of molten steel at the end-point of 120 t converter is 1903 K. The ultimate carbon content of molten steel in 120 t converter is 0.03%.

### 4.4. Model Implementation

The numerical program uses explicit finite difference method, which marches forward with time, solving for the carbon content at next time step by using the input parameters calculated in the previous time step. In each differential time period, the dynamic parameters such as oxygen lance height and molten steel temperature can be roughly regarded as unchanged.

The continuous prediction model of carbon content is realized by programming language C++, which can be used to predict the carbon content during the whole BOF steelmaking process. Figure 5 demonstrates the flowchart of the computation program of the complete mathematical model. When the calculation program starts running, the parameters such as hot metal weight, the temperature of hot metal, hot metal composition, scrap weight, oxygen blowing amount, and oxygen blowing process are given as initial

inputs to the computational program. The oxygen blowing process, including oxygen lance height and oxygen flow rate, is given as input to the model at each time step. In this paper, the step size is set to 1 s. The decarburization rate and desiliconization rate are calculated at each time step. The carbon content and Si content calculated in the previous time step are used as the input parameters of the current step. The duration of the first stage of decarburization is judged by the Si content. The carbon content of molten steel is the condition to judge whether it enters the third stage of decarburization.

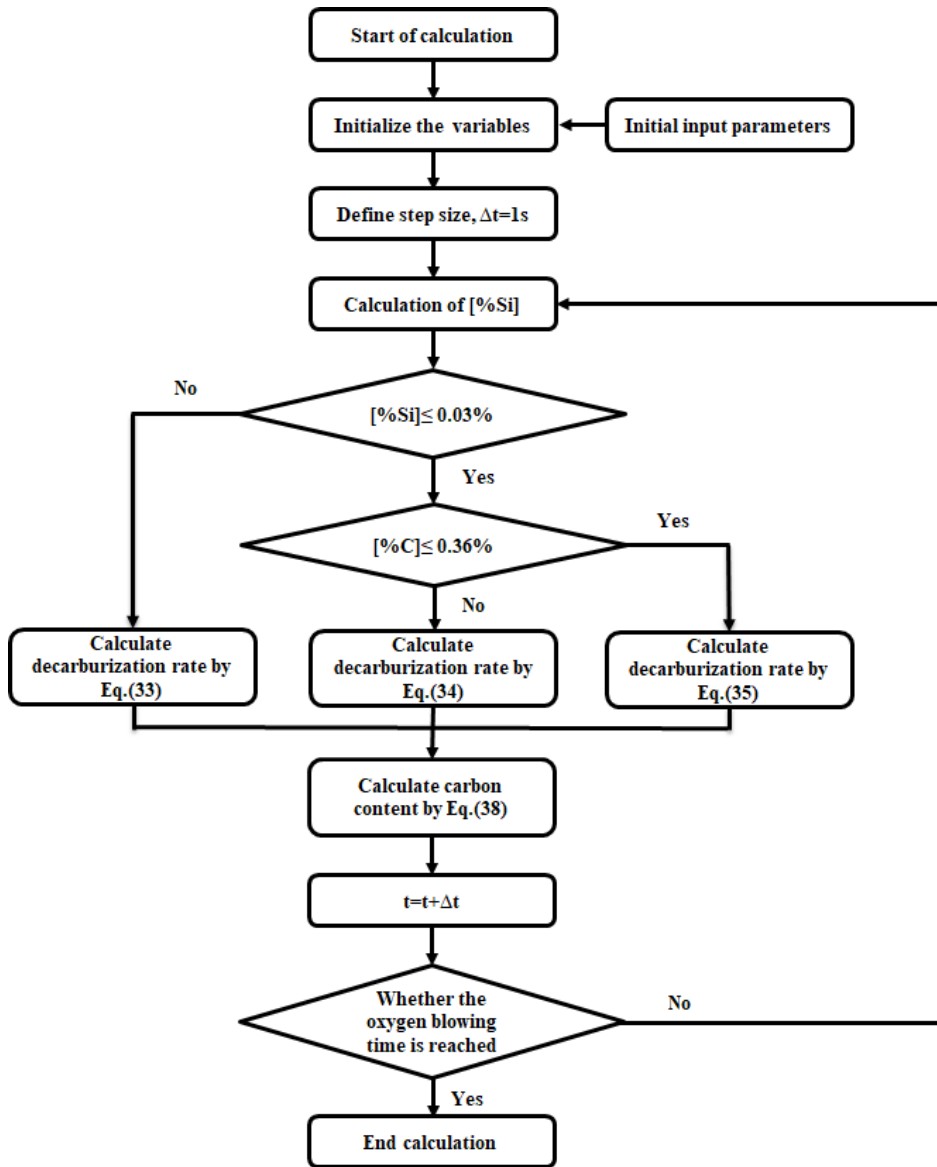

**Figure 5.** Calculation flow chart of the continuous prediction model of carbon content.

According to the explicit finite difference method, the carbon content at $(t + \Delta t)$ is calculated by Equations (33)–(38).

The decarburization rate can be expressed as

$$-\frac{dC}{dt} = \beta k_1 W_{steel} \frac{h_{min}}{h^t} \frac{Q_{top}^t}{Q_{topmax}} t \tag{33}$$

$$-\frac{dC}{dt} = \alpha \beta W_{steel} \frac{h_{min}}{h^t} \frac{Q_{top}^t}{Q_{topmax}} v_{Oxygen}^t \tag{34}$$

$$-\frac{dC}{dt} = \beta k_3 W_{steel} \frac{h_{min}}{h^t} \frac{Q_{top}^t}{Q_{topmax}} \left( [C\%]^t - C_0 \right) \tag{35}$$

The desiliconization rate can be expressed as

$$-\frac{dSi}{dt} = A_{iz}^t k_{gm} \rho_m \left( [Si\%]^t - [Si\%]_{sm}^t \right) / 100 + A_{sm}^t k_{sm} \rho_m \left( [Si\%]^t - [Si\%]_{sm}^t \right) / 100 \tag{36}$$

The carbon content and Si content at $(t + \Delta t)$ can be expressed as

$$[Si\%]^{t+\Delta t} = [Si\%]^t - \frac{dSi}{dt} \Delta t \tag{37}$$

$$[C\%]^{t+\Delta t} = [C\%]^t - \frac{dC}{dt} \Delta t \tag{38}$$

## 5. Model Validation and Discussion

### 5.1. Model Verification

To verify the continuous prediction model of carbon content, 210 heats of actual production data from a 120 t BOF of an iron and steel enterprise in China were collected, of which 150 heats of data were used to determine the coefficients of the prediction model of carbon content, and process verification and end-point verification for the continuous prediction model of carbon content were carried out by the other 60 heats of data. The aim of process verification is to verify whether the decarburization behavior described by the prediction model accords with the actual behavior.

It is assumed that in the second stage of decarburization, CO accounts for 90% and $CO_2$ accounts for 10% of the deoxidation products produced by carbon oxidation reaction. According to this assumption, $\alpha$ can be calculated as 0.00000568. When the carbon content is 0.36%, the decarburization rates of the second decarburization stage and the third decarburization stage are equal.

$k_3$ can be calculated as

$$k_3 = \frac{\alpha v_{Oxygen}^t}{\left( [C\%]^t - C_0 \right)} \tag{39}$$

$k_3$ can be calculated as 0.00023845. According to the carbon content test results and test time of TSC and TSO of 150 heats of data, $\beta k_3$ can be calculated as 0.0002336. $\beta$ can be calculated as 0.98. According to the statistics of 150 heats of data, the average Si content of hot metal is 0.43%. The Si content in the blowing process is calculated according to Equation (20), and the duration in the early stage of decarburization is determined as 349 s. At the end of the first stage, the decarburization rates of the second decarburization stage and the first decarburization stage are equal. $k_1$ can be calculated as $2.25 \times 10^{-7}$.

The choice of turning point between the three stages will affect the duration of each stage and the coefficient of the model. Taking the Si content of hot metal as 0.43%, the critical Si content in the first and second stages are 0.03%, 0.04%, 0.05%, and 0.06% respectively. The first stage duration, coefficient $k_1$, and end-point carbon content calculated by the model are shown in Table 2. When the critical silicon content increases, the duration of the first stage decreases, and the duration of the second stage increases, resulting in the decrease of the end-point carbon content.

The critical carbon content of the second and third stages is 0.36%, 0.26%, and 0.46% respectively. The model calculation results and coefficient $k_3$ are shown in Table 3. The selected critical carbon content increases, the time to enter the third stage of decarburization is advanced, the total decarburization rate is reduced, and the end carbon content increases.

**Table 2.** The effect of critical silicon content on model performance.

| Critical Si Content/% | The First Stage Duration/s | $k_1$ | End-Point Carbon Content/wt% |
|:---:|:---:|:---:|:---:|
| 0.03 | 349 | $2.25 \times 10^{-7}$ | 0.104558 |
| 0.04 | 311 | $2.53 \times 10^{-7}$ | 0.082771 |
| 0.05 | 281 | $2.8 \times 10^{-7}$ | 0.070846 |
| 0.06 | 256 | $3.07 \times 10^{-7}$ | 0.063065 |

**Table 3.** The effect of critical carbon content on model performance.

| Critical C Content/% | $k_3$ | End-Point Carbon Content/wt% |
|:---:|:---:|:---:|
| 0.26 | 0.00034212 | 0.074941 |
| 0.36 | 0.00023845 | 0.104558 |
| 0.46 | 0.000183 | 0.130607 |

The 4 heats of actual production data of No.2131001445, No.2131005122, No.2132003381, and No.2132003685 were used for process validation of the prediction model. The model calculation results and actual test results of the four heats are shown in Figure 6. Among them, the two actual carbon content detection points of each heat are the carbon content detected by TSC and TSO respectively. As shown in Figure 6, the calculated results of the continuous prediction model are in good agreement with the actual measured values. The absolute error between the carbon content calculated by the model and the carbon content detected by TSC is less than 0.03% for No.2131001445, No.2131005122, and No.2132003685. The absolute error between the carbon content calculated by the model and the carbon content detected by TSC is less than 0.042% for No.2132003381. The absolute error between the carbon content calculated by the model and the carbon content detected by TSO is less than 0.01% for No.2131001445, No.2131005122, No.2132003381, and No.2132003685. The results of process verification indicate that the continuous prediction model of carbon content established in the paper basically accords with the actual behavior of decarburization.

The model accuracy verification results are shown in Figure 7. The steel selected in this paper is low carbon steel, and the target carbon content of the converter end-point is 0.06%. As shown in Figure 7, when the end-point carbon content is 0.038–0.1024%, the accuracy of the model can reach 85%, and the absolute error is less than or equal to 0.02%. When the end-point carbon content is 0.045–0.07 wt%, the accuracy of the model can reach 95.92%, and the absolute error is less than or equal to 0.02 wt%. The prediction accuracy can reach 95%, while the absolute error is less than 0.035 wt%. The continuous prediction model of carbon content established in the paper has good prediction accuracy.

*5.2. Discussion*

Both Si content in hot metal and oxygen blowing process affect the decarburization effect of converter. The Si content in hot metal will affect the duration of the first stage of decarburization. Oxygen blowing process will affect the decarburization rate. In order to study the effect of Si content in hot metal and oxygen lance height on decarburization of converter, a specific production heat was selected. The specific information of the production heat is shown in Table 4. The continuous prediction model of carbon content established in the paper was used to calculate this furnace. The end-point carbon content calculated by the model is 0.0685%, the absolute error is 0.0113%.

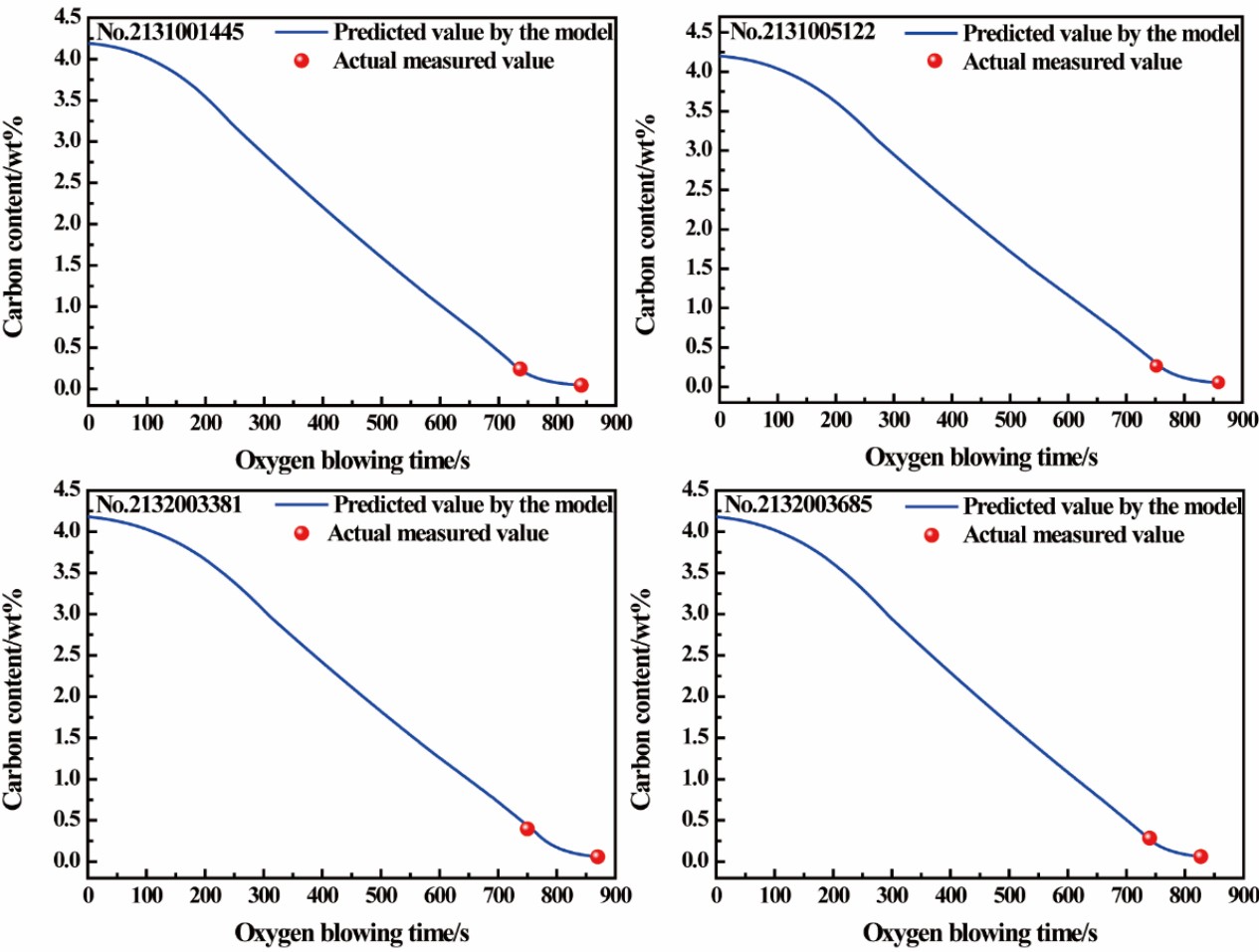

**Figure 6.** Comparison between predicted carbon content and actual measurement values.

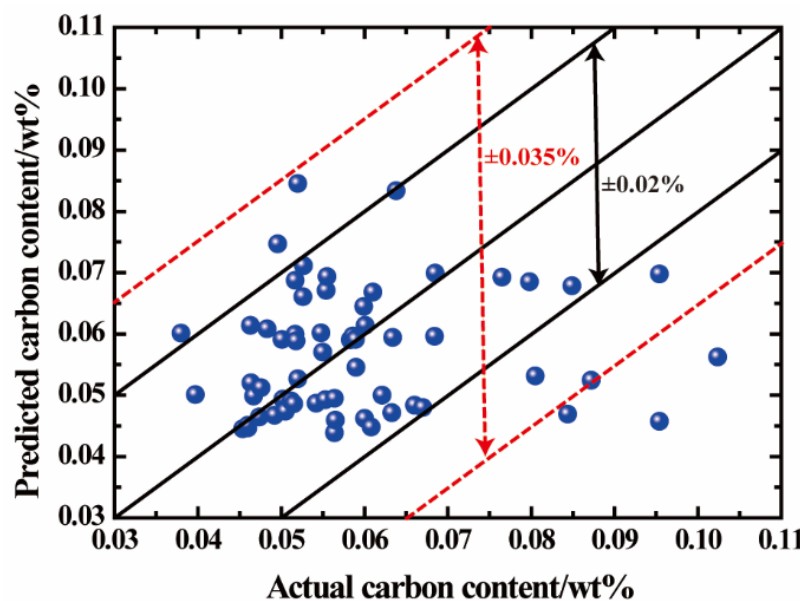

**Figure 7.** The prediction results of carbon content at the steelmaking end-point.

**Table 4.** The specific information of the production heat.

| Parameter | Value |
|---|---|
| Hot metal weight/t | 122 |
| Scrap weight/t | 28.8 |
| Hot metal temperature/°C | 1363 |
| C content/wt% | 4.15 |
| Si content/wt% | 0.61 |
| Blowing oxygen volume/Nm$^3$ | 6599 |

Figure 8 shows the effect of Si content in hot metal on decarburization effect of converter. With the decrease of Si content in hot metal, the duration of the first stage of decarburization decreases significantly, and the carbon content at the end of the first stage increases. Due to the decrease of Si content, the duration of the second stage of decarburization is prolonged and the decarburization efficiency is improved, and the endpoint carbon content is also reduced. Other conditions remain unchanged, the Si content in hot metal is reduced by 0.1%, and the carbon content of molten steel at the end point is relatively reduced by about 11.7%.

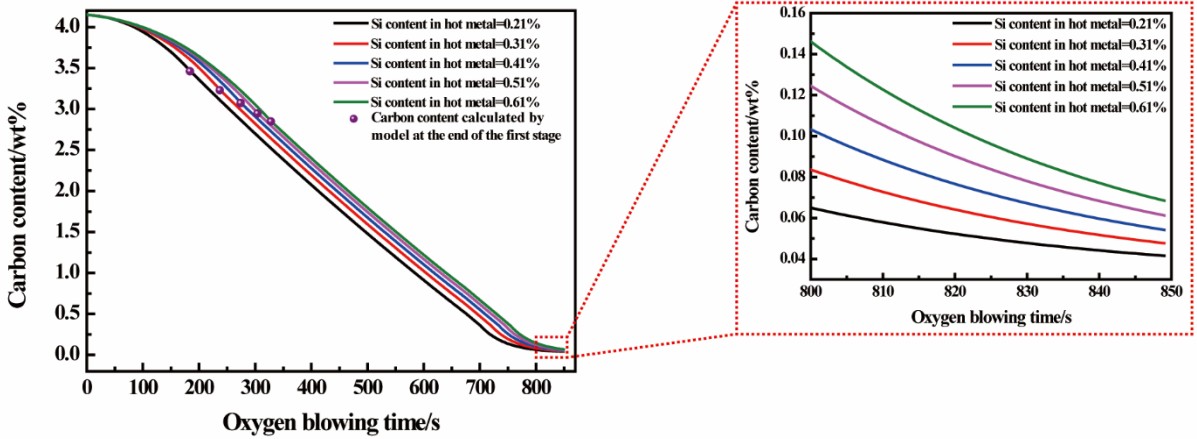

**Figure 8.** The effect of Si content in hot metal on decarburization effect of converter.

In order to study the influence of oxygen lance height on the decarburization process of converter, the oxygen lance height scheme shown in Table 5 was designed, in which programme 2 was the oxygen lance height of actual smelting. Programmes 1–4 are the influence of main blowing oxygen lance height on decarburization process.

**Table 5.** The oxygen lance height schemes (m).

| Oxygen Step | 0 | 0.01 | 0.1 | 0.18 | 0.22 | 0.8 | 0.9 | 1.0 |
|---|---|---|---|---|---|---|---|---|
| programme 1 | 2.14 | 1.9 | 1.8 | 1.7 | 1.65 | 1.5 | 1.5 | 1.5 |
| programme 2 | 2.14 | 1.9 | 1.8 | 1.7 | 1.60 | 1.5 | 1.5 | 1.5 |
| programme 3 | 2.14 | 1.9 | 1.8 | 1.7 | 1.55 | 1.5 | 1.5 | 1.5 |
| programme 4 | 2.14 | 1.9 | 1.8 | 1.7 | 1.50 | 1.5 | 1.5 | 1.5 |
| programme 5 | 2.14 | 1.9 | 1.8 | 1.7 | 1.60 | 1.60 | 1.60 | 1.60 |
| programme 6 | 2.14 | 1.9 | 1.8 | 1.7 | 1.60 | 1.65 | 1.65 | 1.65 |
| programme 7 | 2.14 | 1.9 | 1.8 | 1.7 | 1.60 | 1.70 | 1.70 | 1.70 |

As shown in Figure 9, reducing the main blowing oxygen lance height can improve the decarburization rate in the second stage, and the decarburization efficiency is significantly increased. The main blowing oxygen lance height is reduced by 0.05 m, and the end-point carbon content is relatively reduced by about 11.6%. Programmes 5–7 are the influence of oxygen lance height in the later stage on decarburization process. As shown in Figure 10, the oxygen lance height in later stage is reduced by 0.05 m, and the end-point carbon content is relatively reduced by about 6.86%.

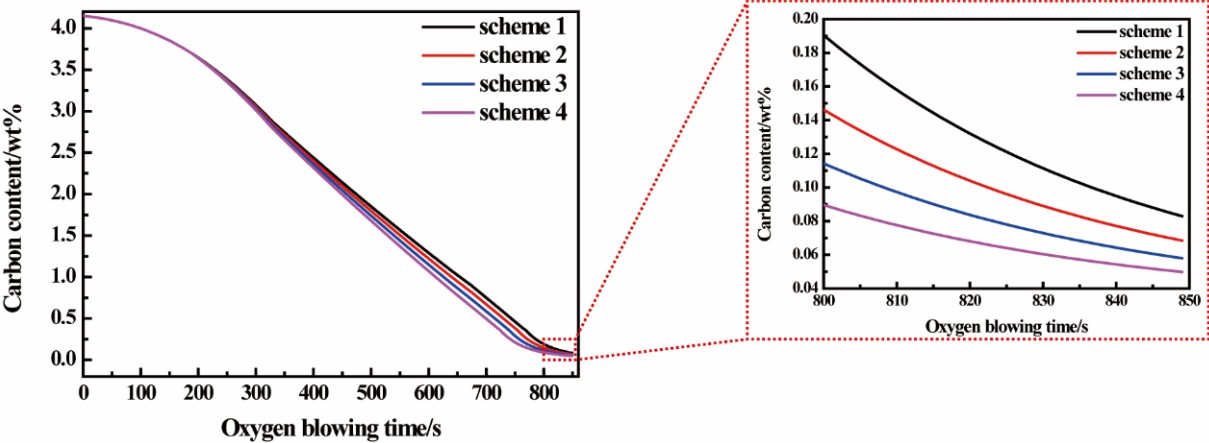

**Figure 9.** The influence of main blowing oxygen lance height on decarburization process.

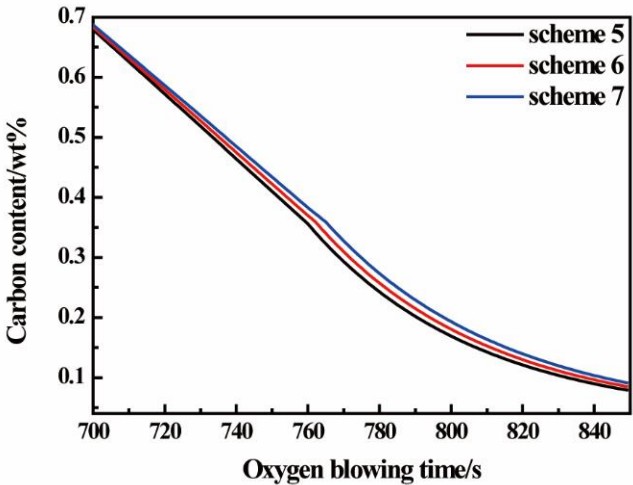

**Figure 10.** The influence of oxygen lance height in later stage on decarburization process.

## 6. Conclusions

Based on the three-stage decarburization theory and the production process of 120 t converter, a continuous prediction model of carbon content in molten steel is established in this paper. The operation of the model is realized by displaying the finite difference method. The following conclusions can be obtained:

1. The results of process verification indicate that the prediction model established in the paper basically accords with the actual behavior of decarburization. The absolute error between the carbon content calculated by the model and the carbon content detected by TSC is within 0.042%.
2. When the end-point carbon content is 0.038–0.1024%, the accuracy of the carbon content predicted by the model can reach 85%, when the end carbon content is 0.045–0.07%, the accuracy of the carbon content predicted by the model can reach 95.92%, and the absolute error is less than or equal to 0.02%. The carbon content prediction model established in this paper has good prediction accuracy.

3. The model is applied to calculate the effects of Si content in hot metal and oxygen lance height on the end-point carbon content. The results show that the Si content in hot metal decreases by 1%, and the end-point carbon content is relatively reduced by 11.7%, The main blowing oxygen lance height and later oxygen lance height decreased by 0.05 m respectively, and the end-point carbon content is relatively reduced by about 11.6% and 6.86%.

**Author Contributions:** Investigation, F.G. and J.C.; project administration, Y.B.; writing—original draft, D.W.; writing—review and editing, D.W., L.X. and Y.B. All authors have read and agreed to the published version of the manuscript.

**Funding:** This research received no external funding.

**Institutional Review Board Statement:** Not applicable.

**Informed Consent Statement:** Not applicable.

**Data Availability Statement:** All data and models generated or used during the study appear in the submitted article.

**Conflicts of Interest:** The authors declare no conflict of interest.

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
