# Peer review of "Continuous Prediction Model of Carbon Content in 120 t Converter Blowing Process"

_metals, doi:10.3390/met12010151_

Round 1

Reviewer 1 Report

The paper describes a simple model of a steelmaking converter, with the aim of predicting the end-point carburization of the steel. The model is based on a three-stage division of the blowing process and introduces certain assumption concerning when the stages end.

While the overall approach is interesting, and the simplicity of the model is appreciated, as it will mean that the effect of the parameters can be easily and rapidly analyzed, certain key questions arise when the paper was reviewed:

Major issues.

The simplified equations are introduced by setting certain threshold values, where one stage ends and the next starts. These conditions are crucial for the performance of the model. How much do these assumptions (Si=0.03%, C=0.36%) affect the performance of the model, and are these numbers heavily dependent on the grade of the steel produced?

In many of the expressions, there are products of unknown parameters, which means that it is challenging, or even impossible, to estimate their values appropriately. How were the parameters estimated in practice, and can unique values for them really be found? 

The justification for assuming a linear dependence of the lance height and dC/dt is not clear. Often it is claimed that it is the distance between the lance tip and the iron/steel bath surface this is important. How should the model be adjusted if the batch size in the converter changes, or if the bath level changes under the blow?

The surface areas of slag and iron (Eq. (19)) are important for the results. How were these estimated as they are largely unknown? Also in these equations, the authors include products of unknown parameters, which makes a decent parameter estimation complicated. 

Are the discrete points in Fig. 8 properly placed in time. If so, why is there so much of mismatch?

Minor issues:

The values and units should be separated by a space, e.g., 120 t ad not 120t. Please check also other values with units in the text.  

The caption of Figure 1 is odd: does the figure really present “traditional theory”?

It is not appropriate to report number with four digits (e.g., 74.92%) as done in Section 3.3 for intervals that have been rather arbitrarily chosen, and the decimals anyway are completely irrelevant. I suggest you round all these number to integers.

In mathematical equationa, the cross for multiplication should be omitted, as it is not normally used (or needed).

Why is W_{hotel} divided by 100? This scaling can as well be absorbed in the parameters (e.g., k_1, k_2,…). Also, the subscript “hotel” is very misleading. You may use, e.g., “steel” or something similar, as the quantity has nothing to do with hotels.

In Eq. (9), why isn’t there a lower index i on Ea as well, or isn’t this reaction dependent?

After an equation, where it is part of a sentence that continues, the “where” that follows in defining the symbols should neither be indented nor capitalized.  If you start a sentence with “Where…” it is a question!

On line 318: How do you define the concept “decarburization efficiency”?

There is a double numbering of the items in the conclusions.

Author Response

Responses to Comments

Dear Editor and Referees,

On behalf of my co-authors, we thank you very much for giving us an opportunity to revise our manuscript, we appreciate you and reviewers very much for your positive and constructive comments and suggestions on our manuscript entitled “Continuous prediction model of carbon content in 120 t converter blowing process”. (Manuscript ID: metals-1504115). We have studied reviewer’s comments carefully and have made revision which are highlighted in red in the revised manuscript. Meanwhile, we responded point by point to all the comments as listed below. Attached please find the revised version, which we would like to submit for your kind consideration.

Reply to the reviewer’s comments:

To referee 1:

  1. Comment: The simplified equations are introduced by setting certain threshold values, where one stage ends and the next starts. These conditions are crucial for the performance of the model. How much do these assumptions (Si=0.03%, C=0.36%) affect the performance of the model, and are these numbers heavily dependent on the grade of the steel produced?

Reply: Thanks for your nice comments. The critical silicon content and critical carbon content mainly depend on the content of impurity elements in the converter, oxidation reaction progress, molten pool temperature and liquid steel stirring, and have little relationship with the steel grade. The influence of critical silicon content and critical carbon content on the performance of the model is added to the manuscript.

Please see the following.

“The choice of turning point between the three stages will affect the duration of each stage and the coefficient of the model. Taking the Si content of hot metal as 0.43%, the critical Si content in the first and second stages are 0.03%, 0.04%, 0.05% and 0.06% respectively. The first stage duration, coefficient k1 and end-point carbon content calculated by the model are shown in Table 2. When the critical silicon content increases, the duration of the first stage decreases, and the duration of the second stage increases, resulting in the decrease of the end-point carbon content

Table 2. The effect of critical silicon content on model performance.

Critical Si content/%

The first stage duration/s

k1

End-point carbon content/ wt%

0.03

349

2.25×10-7

0.104558

0.04

311

2.53×10-7

0.082771

0.05

281

2.8×10-7

0.070846

0.06

256

3.07×10-7

0.063065

The critical carbon content of the second and third stages is 0.36%, 0.26% and 0.46% respectively. The model calculation results and coefficient k3 are shown in Table 3. The selected critical carbon content increases, the time to enter the third stage of decarburization is advanced, the total decarburization rate is reduced, and the end carbon content increases

Table 3. The effect of critical carbon content on model performance.

Critical C content/%

k3

End-point carbon content/ wt%

0.26

0.00034212

0.074941

0.36

0.00023845

0.104558

0.46

0.000183

0.130607

(Please see line 339-353 of Page 7, red-labeled part).

  1. Comment: In many of the expressions, there are products of unknown parameters, which means that it is challenging, or even impossible, to estimate their values appropriately. How were the parameters estimated in practice, and can unique values for them really be found?

Reply: Thanks for your valuable comments. Through reasonable assumptions, statistical analysis of the actual production data and the relationship between the three decarbonization stages, the unknown parameters in the model calculation formula can be calculated. The specific calculation process has been added to the manuscript.

Please see the following.

“It is assumed that in the second stage of decarburization, CO accounts for 90% and CO2 accounts for 10% of the deoxidation products produced by carbon oxidation reaction. According to this assumption, α can be calculated as 0.00000568. When the carbon content is 0.36%, the decarburization rates of the second decarburization stage and the third decarburization stage are equal.

k3 can be calculated as

     (39)

k3 can be calculated as 0.00023845. According to the carbon content test results and test time of TSC and TSO of 150 heats of data, βk3 can be calculated as 0.0002336. β can be calculated as 0.98. According to the statistics of 150 heats of data, the average Si content of hot metal is 0.43%. The Si content in the blowing process is calculated according to equation (20), and the duration in the early stage of decarburization is determined as 349 s. At the end of the first stage, the decarburization rates of the second decarburization stage and the first decarburization stage are equal. k1 can be calculated as 2.25×10-7. ”

(Please see line 325-338 of Page 10, red-labeled part).

  1. Comment: How should the model be adjusted if the batch size in the converter changes, or if the bath level changes under the blow?

Reply: Thanks for your nice comments. With the increase of converter life, the size of converter molten pool will change, especially the height of molten pool. In this case, the accurate position of converter lance position also needs to be determined again. Usually, the staff of the steelmaking plant will determine the lance position of the converter after smelting for a certain time. After re determining the lance position, it is necessary to collect the latest production data and adjust the parameters (α, β, k1, k3) of the model to ensure the accuracy of the model.

  1. Comment: The surface areas of slag and iron (Eq. (19)) are important for the results. How were these estimated as they are largely unknown? Also in these equations, the authors include products of unknown parameters, which makes a decent parameter estimation complicated.

Reply: Thanks for your valuable comments. The calculation process of the interfacial area of gas-metal reaction and the interfacial area of slag-metal reaction are added to the manuscript.

Please see the following.

“The interfacial area of gas-metal reaction can be expressed as

     (21)

where nn is the number of nozzles; riz is the radius of the gas-metal reaction zone, m2; hiz is the height of the gas-metal reaction zone, m. hiz and riz can be expressed as

     (22)

      (23)

      (24)

      (25)

     (26)

     (27)

where dth is the throat diameter of the lance, m; Pa is the ambient pressure, Pa; P0 is the top supply pressure, Pa; mt is the total momentum flow rate; mn is the momentum flow rate of the each nozzle; Mh and Md are the dimensionless momentum flow rate.

The interfacial area of slag-metal reaction can be expressed as

      (28)

where Db is the diameter of the bath surface, m.”

(Please see line 227-244 of Page 6, red-labeled part).

The determination method of unknown parameters has been mentioned in reply 1.

  1. Comment: Are the discrete points in Fig. 8 properly placed in time. If so, why is there so much of mismatch?

Reply: Thanks for your nice comments. The discrete points in Figure 8 are the carbon content at the end of the early stage of decarburization calculated by the model in each decarburization curve. In Fig. 8, the identification of discrete points has been modified to“Carbon content calculated by model at the end of the first stage”.

Figure 8. The effect of Si content in hot metal on decarburization effect of converter.

  1. Comment: The values and units should be separated by a space, e.g., 120 t ad not 120t. Please check also other values with units in the text.

Reply: Thanks for your valuable comments. We have checked all values with units in the manuscript. Values and units are separated by spaces.

  1. Comment: The caption of Figure 1 is odd: does the figure really present “traditional theory”?

Reply: Thanks for your nice comments. We have modified Figure 1.  In the original figure, the curve appears too high due to the improper value of ordinate.

Figure 1. Traditional three-stage decarburization theory for BOF process.

  1. Comment: It is not appropriate to report number with four digits (e.g., 74.92%) as done in Section 3.3 for intervals that have been rather arbitrarily chosen, and the decimals anyway are completely irrelevant. I suggest you round all these number to integers.

Reply: Thanks for your nice comments. We have rounded all four digits (e.g. 74.92%) in Section 3.3 to integers.

  1. Comment: In mathematical equationa, the cross for multiplication should be omitted, as it is not normally used (or needed).

Reply: Thanks for your valuable comments. The unnecessary cross for multiplication in mathematical equations has been omitted.

  1. Comment: Why is W_{hotel} divided by 100? This scaling can as well be absorbed in the parameters (e.g., k_1, k_2,…). Also, the subscript “hotel” is very misleading. You may use, e.g., “steel” or something similar, as the quantity has nothing to do with hotels.

Reply: Thanks for your nice comments. We have changed Whotel to Wsteel. And the scaling has been absorbed in the parameters ki.

  1. Comment: In Eq. (9), why isn’t there a lower index i on Ea as well, or isn’t this reaction dependent?

Reply: Thanks for your valuable comments. Ea is the activation energy of the reaction. Ea is related to the reaction temperature and is not a constant in stages. So there isn’t a lower index i on Ea.

  1. Comment: After an equation, where it is part of a sentence that continues, the “where” that follows in defining the symbols should neither be indented nor capitalized. If you start a sentence with “Where…” it is a question!

Reply: Thanks for your nice comments. We've checked the “where” after each equation. "where" is neither indented nor capitalized

  1. Comment: How do you define the concept “decarburization efficiency”?

Reply: Thanks for your valuable comments. "Decarburization efficiency is the ability to remove the carbon content under the same oxygen blowing process. Due to the reduction of the first stage of decarburization, more carbon can be removed under the premise of the same oxygen blowing.

  1. Comment: There is a double numbering of the items in the conclusions.

Reply: Thanks for your valuable comments. We have revised the double number in the conclusion item.

Reviewer 2 Report

The dynamic model of carbon oxidation in the converter process requires the use of information collected during the oxygen blow. The proposed model takes into account only the height of the lance position and the intensity of the oxygen blow. This type of model seems to be insufficient for cases deviating from standard conditions.

The essence of the proposed solution are equations 6 - 8. These equations include, among others, parameters k1, k3 and β which require verification. The authors do not provide a method for verifying these parameters. Without this knowledge, it is impossible to assess whether the proposed model can be used for quantitative calculations. For this reason, the article needs to be completely supplemented.

The above remark is fundamental. In addition, there are a few more statements in the article that are quite controversial

  1. Line 38 - from an industrial point of view, the statement that " the prediction and control of endpoint carbon content is more important than that of endpoint temperature" is not true
  2. The assumption that the critical content of carbon separating the second and third phases of the blow has a constant value is also not confirmed in industrial practice.

For the reasons described above, the article requires thorough improvement.

Author Response

Responses to Comments

Dear Editor and Referees,

On behalf of my co-authors, we thank you very much for giving us an opportunity to revise our manuscript, we appreciate you and reviewers very much for your positive and constructive comments and suggestions on our manuscript entitled “Continuous prediction model of carbon content in 120 t converter blowing process”. (Manuscript ID: metals-1504115). We have studied reviewer’s comments carefully and have made revision which are highlighted in red in the revised manuscript. Meanwhile, we responded point by point to all the comments as listed below. Attached please find the revised version, which we would like to submit for your kind consideration.

Reply to the reviewer’s comments:

To referee 2:

  1. Comment: The dynamic model of carbon oxidation in the converter process requires the use of information collected during the oxygen blow. The proposed model takes into account only the height of the lance position and the intensity of the oxygen blow. This type of model seems to be insufficient for cases deviating from standard conditions.

Reply: Thanks for your nice comments. The dynamic model needs process detection information to dynamically adjust the operation process. The model established in this paper can be regarded as a static model of mechanism. This model does not need dynamic detection information in the calculation process. According to the set oxygen blowing amount and oxygen blowing process, the end-point carbon content can be predicted. The blowing oxygen can be adjusted according to the calculation result.

  1. Comment: The essence of the proposed solution are equations 6 - 8. These equations include, among others, parameters k1, k3 and β which require verification. The authors do not provide a method for verifying these parameters. Without this knowledge, it is impossible to assess whether the proposed model can be used for quantitative calculations. For this reason, the article needs to be completely supplemented.

Reply: Thanks for your valuable comments. Through reasonable assumptions, statistical analysis of the actual production data and the relationship between the three decarbonization stages, the unknown parameters in the model calculation formula can be calculated. The specific calculation process has been added to the manuscript.

Please see the following.

“It is assumed that in the second stage of decarburization, CO accounts for 90% and CO2 accounts for 10% of the deoxidation products produced by carbon oxidation reaction. According to this assumption, α can be calculated as 0.00000568. When the carbon content is 0.36%, the decarburization rates of the second decarburization stage and the third decarburization stage are equal.

k3 can be calculated as

     (39)

k3 can be calculated as 0.00023845. According to the carbon content test results and test time of TSC and TSO of 150 heats of data, βk3 can be calculated as 0.0002336. β can be calculated as 0.98. According to the statistics of 150 heats of data, the average Si content of hot metal is 0.43%. The Si content in the blowing process is calculated according to equation (20), and the duration in the early stage of decarburization is determined as 349 s. At the end of the first stage, the decarburization rates of the second decarburization stage and the first decarburization stage are equal. k1 can be calculated as 2.25×10-7.”

(Please see line 325-338 of Page 10, red-labeled part).

  1. Comment: Line 38 - from an industrial point of view, the statement that " the prediction and control of endpoint carbon content is more important than that of endpoint temperature" is not true

Reply: Thanks for your nice comments. This statement is indeed incorrect. We have modified this expression as“Due to the fast decarburization speed in converter and the narrow range of carbon content required by steel grade specification, it is very difficult to predict and control the end-point carbon content.”

(Please see line 37-39 of Page 1, red-labeled part).

  1. Comment: The assumption that the critical content of carbon separating the second and third phases of the blow has a constant value is also not confirmed in industrial practice.

Reply: Thanks for your valuable comments. The critical carbon content Ct is between 0.2%~0.6%. Generally, the critical carbon content is usually related to the stirring strength of the molten pool. In some literatures, the critical carbon content is selected for convenience calculation. We have modified this expression as“Generally, the critical carbon content is usually related to the stirring strength of the molten pool. The critical carbon content Ct is between 0.2%~0.6%.[26] Some scholars choose the critical carbon content as 0.35% or 0.43%.[27]”

(Please see line 255-258 of Page 7, red-labeled part).

Reviewer 3 Report

Review Report

1) A brief summary outlining the aim of the paper and its main contributions:

The main aim of the manuscript is to present the evolution of dynamic model of decarburization during steel production in basic oxygen furnace.

Because the production of steel in BOF is still the most common method of the primary production of steel, it could be interesting for readers.

But from my point of view, the many more complex papers were already published on these topics which consider not only the decarburization (end point of decarburization) but all reactions and processes accompanying the production steel in basic oxygen convertors (for example: A. K. Shukla,B. Deo,S. Millman,B. Snoeijer,A. Overbosch,A. Kapilashrami: An Insight into the Mechanism and Kinetics of Reactions In BOF Steelmaking: Theory vs Practice. steel research int. 81 (2010) No. 11, p.940-948).

2) Therefore, on the basis of the above-mentioned, I see shortcomings especially (Specific comments):

Introduction: insufficient declaration in which part the contribution is new compared to other authors

Chap.2: Insufficient description of the reactions taking place in BOF and their influence on the course of the carbon reaction. Method of carbon oxidation ( up to 85% of the accompanying elements in hot metal are oxidised by oxygen dissolved in the melt - I lack a description of the method of oxidation of the accompanying elements - gaseous oxygen, atomically dissolved oxygen and oxygen bound to FeO). In practice, the blowing is finished no according the carbon content but according the final content of phosphorus – please, also discuss this problem in your manuscript.

Chap.3, Line 97, wrong formulation: And the hot metal temperature of 74.92% of the heats is concentrated at 4.15%~4.25%.

Chap. 3, Line 111-112: specify and state that these are the final carbon and steel temperature values achieved after oxygen blowing.

Equations: change the index in “Whotel” on “w steel”?

Line 214, Eq. 12: Where did the -5168/T values come from? This is probably the temperature dependence of the equilibrium constant. I lack explanation and clarification.

Line 300: The description does not correspond to Figure 7 - Figure 7 does not indicate the accuracy of the model.

No other technical suggestions.

Review does not comment on formatting issues that do not obscure the meaning of the paper, as these will be addressed by editors.

Reviewer

Author Response

Responses to Comments

Dear Editor and Referees,

On behalf of my co-authors, we thank you very much for giving us an opportunity to revise our manuscript, we appreciate you and reviewers very much for your positive and constructive comments and suggestions on our manuscript entitled “Continuous prediction model of carbon content in 120 t converter blowing process”. (Manuscript ID: metals-1504115). We have studied reviewer’s comments carefully and have made revision which are highlighted in red in the revised manuscript. Meanwhile, we responded point by point to all the comments as listed below. Attached please find the revised version, which we would like to submit for your kind consideration.

Reply to the reviewer’s comments:

To referee 3:

  1. Comment: Introduction: insufficient declaration in which part the contribution is new compared to other authors

Reply: Thanks for your nice comments. This part of information is supplemented in the introduction.

“The model parameters such as ultimate carbon content of molten steel are calculated according to the enterprise data. And the dynamic transmission of process parameters is realized by finite difference method. This model only needs raw material information and process information to run without additional testing information. This model is more practical for many iron and steel enterprises lacking testing equipment.”

(Please see line 64-69 of Page 2, red-labeled part).

  1. Comment: Insufficient description of the reactions taking place in BOF and their influence on the course of the carbon reaction. Method of carbon oxidation ( up to 85% of the accompanying elements in hot metal are oxidised by oxygen dissolved in the melt - I lack a description of the method of oxidation of the accompanying elements - gaseous oxygen, atomically dissolved oxygen and oxygen bound to FeO). In practice, the blowing is finished no according the carbon content but according the final content of phosphorus – please, also discuss this problem in your manuscript.

Reply: Thanks for your valuable comments. We have added the description of the reactions taking place in BOF and their influence on the course of the carbon reaction.

“In addition to the oxidation of carbon, the following reactions occur in the converter. The oxidation of Si, Mn and Fe as shown in equation (4) ~ (6) occurs in the oxygen jet reaction zone. The reaction shown in equation (7) ~ (10) occurs in the slag-metal interface reaction zone. The reaction shown in equation (3) occurs in the emulsion phase reaction zone.

     (3)

     (4)

     (5)

     (6)

     (7)

     (8)

     (9)

     (10)

     (11)

From the thermodynamic point of view, the ability of many chemical elements in the molten pool, especially silicon and manganese, to combine with oxygen is much greater than that of carbon. When w(Si)+0.25 w(Mn)<0.1%, the content of silicon and manganese has little effect on the rate of carbon oxygen reaction.”

(Please see line 80-97 of Page 2 and 3, red-labeled part).

And we have discussed the problem of the blowing is finished no according the carbon content but according the final content of phosphorus.

“Generally, three conditions are required for the converter to finish smelting. The phosphorus content of molten steel is lower than the requirements of steel grade. The carbon content and temperature of molten steel meet the process requirements. Since it is difficult to reduce the phosphorus content of molten steel in the subsequent refining process, the requirements for phosphorus content at the end of converter are more strict. However, the control of end-point carbon content is equally important. Too high carbon content will increase the content of sulfur and phosphorus, affect the desulfurization and dephosphorization operation, and too low carbon content will increase the content of oxygen and nitrogen. The control of end-point carbon content has an important impact on molten steel quality and smelting efficiency.”

(Please see line 109-118 of Page 3, red-labeled part).

  1. Comment: Line 97, wrong formulation: And the hot metal temperature of 74.92% of the heats is concentrated at 4.15%~4.25%.

Reply: Thanks for your nice comments. We have changed “And the hot metal temperature of 74.92% of the heats is concentrated at 4.15%~4.25%.” to “And the carbon content of hot metal of 75% of the heats is concentrated at 4.15%~4.25%.”

(Please see line 128 of Page 3, red-labeled part).

  1. Comment: Chap. 3, Line 111-112: specify and state that these are the final carbon and steel temperature values achieved after oxygen blowing.

Reply: Thanks for your valuable comments. We have stated that “The carbon content and temperature control at the end of oxygen blowing are shown in Figure 4.”

(Please see line 138 of Page 4, red-labeled part).

  1. Comment: Equations: change the index in “Whotel” on “w steel”?

Reply: Thanks for your nice comments. We have changed Whotel to Wsteel.

  1. Comment: Line 214, Eq. 12: Where did the -5168/T values come from? This is probably the temperature dependence of the equilibrium constant. I lack explanation and clarification.

Reply: Thanks for your valuable comments. The -5168/T values come from the relationship between standard Gibbs free energy and equilibrium constant. We have modified the relevant formula.

     (29)

(Please see line 264 of Page 7, red-labeled part).

  1. Comment: Line 300: The description does not correspond to Figure 7 - Figure 7 does not indicate the accuracy of the model.

Reply: Thanks for your nice comments. We have modified Figure 7 so that figure 7 can more clearly show the accuracy of the model. It can be seen from Figure 7 that the calculated data error of 51 heats is within 0.02%, and there are 60 heats in total. So the accuracy of the model can reach 85%, and the absolute error is less than or equal to 0.02%.

Figure 7. The prediction results of carbon content at the steelmaking end-point.

Round 2

Reviewer 1 Report

The authors have addressed the weak points and have provided sufficient explanations. The paper can now be published.

Author Response

We are appreciative for the reviewer's kind recommendation and careful review.

Reviewer 2 Report

After the introduced additions and corrections, the article may be published 

Author Response

(The authors gave the same response as above.)

Reviewer 3 Report

Please, check the equations:

Eq.(4): there is missing the brackets at SiO2

Eq. (7): there is missing the nr. 5 before [Fe]

Author Response

Responses to Comments

Dear Editor and Referees,

On behalf of my co-authors, we thank you very much for giving us an opportunity to revise our manuscript, we appreciate you and reviewers very much for your positive and constructive comments and suggestions on our manuscript entitled “Continuous prediction model of carbon content in 120 t converter blowing process”. (Manuscript ID: metals-1504115). We have studied reviewer’s comments carefully and have made revision which are highlighted in red in the revised manuscript. Meanwhile, we responded point by point to all the comments as listed below. Attached please find the revised version, which we would like to submit for your kind consideration.

Reply to the reviewer’s comments:

To referee 3:

  1. Comment: Eq.(4): there is missing the brackets at SiO2

Reply: Thanks for your nice comments. We have added parentheses to SiO2 in Eq.(4)

     (4)

(Please see line 86 of Page 2, red-labeled part).

  1. Comment: Eq. (7): there is missing the nr. 5 before [Fe]

Reply: Thanks for your valuable comments. We have modified Eq.(7)

     (7)

(Please see line 89 of Page 2, red-labeled part).
